# Impact of Government Intervention in Response to Coronavirus Disease 2019

**DOI:** 10.3390/ijerph192316070

**Published:** 2022-12-01

**Authors:** Jin-Young Won, Yu-Rim Lee, Myeong-Heum Cho, Yoon-Tae Kim, Bo-Young Heo

**Affiliations:** National Disaster Management Research Institute, Ulsan 44538, Republic of Korea

**Keywords:** quarantine, COVID-19, government policies, SARS-CoV-2

## Abstract

Coronavirus disease 2019 (COVID-19) led to the loss of lives and had serious social and economic effects. Countries implemented various quarantine policies to reduce the effects. The countries were divided into low- and high-risk groups based on the differences in quarantine policies and their levels of infection. Quarantine policies that significantly contributed to risk reduction were determined by analyzing 11 quarantine indicators for reducing the spread of COVID-19. The cross-tabulation and Chi-square tests were used to compare the quarantine policies by the groups. Multivariate logistic regression was used to determine the useful quarantine policies implemented by the low-risk group to verify quarantine policies for minimizing the negative effects. The analysis showed that the low- and medium-risk groups showed significant differences for 9 of the 11 indicators, and 4 of these differentiated the low- from the medium-risk group. Countries with strict quarantine policies related to workplace closure and staying at home were more likely to be included in the low-risk group. These policies had a significant impact in the low-risk countries and could contribute to reducing the spread and effects of COVID-19 in countries included in the high-risk group.

## 1. Introduction

On 31 December 2019, 27 people had pneumonia of unknown cause in Wuhan, Hubei Province, China. The cause was identified as severe acute respiratory syndrome coronavirus 2 (SARS-CoV-2), and the World Health Organization (WHO) named the disease coronavirus disease 2019 (COVID-19) [1,2]. COVID-19 spread from Wuhan across China and, at the same time, to neighboring Asian countries such as Thailand, Japan, Korea, Singapore, and Hong Kong. It rapidly spread to Europe through Italy, where there were frequent economic exchanges with China [3], and the entire globe. In January 2020, the WHO declared COVID-19 as a Public Health Emergency of International Concern (PHEIC).

The traditional public health measures for infectious diseases are isolation, quarantine, social distancing, and community containment, and their main goal is to prevent spread [4]. Social distancing reduces physical interactions among people in communities and may involve wearing a mask, avoiding social gatherings, and refraining from going out and working from home. Containment, which is also called lockdown and is the most extreme measure, restricts traveling and can be enforced when many people are ill due to an epidemic in a specific region or country [5]. During the early stages of the COVID-19 pandemic, a Find, Test, Tracing, Isolation, Support (FTTIS) system was essential to prevent the spread of new infectious diseases that were impossible to treat or vaccinate [6].

Governments implemented various non-pharmaceutical intervention (NPI) policies, including social distancing, quarantine, cordons Sanitaire, and mask policies to mitigate the spread of COVID-19 [7]. Similar policies were implemented in several countries; however, the levels and trends of infection differed markedly. To identify the types of quarantine policies and their effectiveness in reducing the risk of infectious diseases, this study evaluated the quarantine policies according to the level of risk of COVID-19 and their strictness in each country and determined those that were the most effective in reducing risks.

The wearing of a mask became mandatory in each region of Germany. A comparison of the rate of infections in regions with and without the mask policy showed that the number of new infections decreased by an approximate mean of 45% for 20 days after the mandatory mask-wearing policy, ranging from 15% to 75% depending on the regions [8]. In countries with cultural norms or government policies that supported mask wearing in the public, the per capita COVID-19 death rate increased by an average of 16.2% weekly, compared to 61.9% weekly in the rest of the countries without policies. Social norms, government policies that support wearing masks in public, and international travel bans were shown to be independently associated with reduced per capita mortality due to COVID-19 [9].

For social distancing, a physical distance of at least 1 m provided protection against the spread of the virus, but up to 2 m may be more effective, and wearing a face mask can significantly reduce the risk of infection [10]. It was reported that a physical distance of 1 m in medical and community settings reduced the risk by 82%, and the relative protection more than doubled for each additional 1 m of distancing and could be effective for up to 3 m [11].

In the case of the national lockdown policy, an analysis of the trends of confirmed cases and deaths before and after the national lockdown in Spain and Italy showed a significant decrease in the incidence after social distancing enforcement [12]. Thailand implemented an immediate nationwide lockdown policy when the number of confirmed cases was the highest and, the number of confirmed cases decreased [13]. The analysis of the effect of containment and closure policies in 157 countries for 303 days from 23 January 2020, to 20 November 2020, showed that stricter school and workplace closure policies using the Oxford COVID-19 Government Response Tracker from Oxford University, UK, reduced the spread of COVID-19. Considering the heteroscedasticity and autocorrelation between panels, stronger reinforcement of lockdown and closure policies, except those for stay-at-home and movement restrictions within the region, led to a more significant reduction in the prevalence of COVID-19 [14]. In a study using data from Epidemic Forecasting Global NPI (EPGNPI) and OxCGRT, the closure of all educational institutions, restricting gatherings to less than ten people, and closure of in-person workplaces significantly reduced the spread of COVID-19, but the effect of the stay-at-home order was minimal [7].

Regarding the stay-at-home policy, a 1% decrease in the number of outings reduced the incidence by approximately 1% over seven days (a series of intervals). Implementing a strict stay-at-home policy for short periods costs less economically than long-term loose implementations. For example, mobility in Wuhan decreased by 85% after the lockdown, and the incidence was estimated to have decreased by 50% in just 12 days [15]. A study conducted in the UK [16] found that a combination of quarantine and follow-up strategies could reduce transmission better than group testing or self-isolation alone (50–60% versus 2–30%) and combining patient isolation with voluntary isolation for three months could prevent 31% of deaths [17]. 

The INFORM COVID-19 Risk Index, released by the EU in 2020, is a comprehensive index that identifies countries at risk from the health and humanitarian impacts of COVID-19 in need of additional international assistance and enables country-by-country comparisons. The risk index calculation consists of three dimensions, and the elements that make it up are as follows: The components of Hazard & Exposure are Population and WaSH (Water, sanitation and hygiene); Vulnerability is composed of Movement, Behavior, Demographic and Comorbidities and Socio-Economic Vulnerability, and Vulnerable Groups elements; and Lack of coping capacity is composed of Health Capacity, Institutional, and Infrastructure [18]. In other words, the INFORM COVID-19 Risk Index is primarily related to the structured risk factors that existed prior to the outbreak. It can be efficiently used to identify the national priorities of early preparation and response measures for infection diseases; however, it is not judged to be suitable for the purpose of monitoring risks that change as a result of a pandemic such as COVID-19. 

Several studies have evaluated the effects of quarantine policies for infectious diseases, but no study has analyzed quarantine policies that have a significant effect on reducing the effects of infectious diseases and determined priorities according to the level of risk in each country. Therefore, based on this study, we presume that high-risk countries can implement optimal quarantine policies promptly and reduce the spread and effects.

## 2. Materials and Methods

### 2.1. Data Collection

For data on the COVID-19 outbreak trend, the number of new deaths per million population and the number of new confirmed cases per million population by country, provided by the Center for Systems Science and Engineering at Johns Hopkins University (CSSE), were used. 

For data on COVID-19-related quarantine policies, we used the policy response index and governments’ Stringency Index provided by the Oxford COVID-19 Government Response Tracker (OXCGRT) at Oxford University, UK. The OXCGRT collects systematic information on policy measures that governments have taken to tackle COVID-19 and consistently tracks and compares quarantine policies by country. The different policy measures were tracked since 1 January 2020, for more than 180 countries and categorized under 21 indicators, such as school closures, travel restrictions, and vaccination policy. These policies were evaluated on a scale to reflect the extent of government action, and scores were aggregated into a suite of policy indices. The policy indicators included eight on lockdown and closure policies (school closure, movement restriction, etc.), two on economic policies (income support, debt relief for households), seven on health system policies (testing policy, contact tracing, etc.), and four on vaccine policies (vaccine prioritization, etc.). 

In this study, eight indicators related to lockdown and closure policies that directly contributed to reducing the spread of infection and three indicators on health system policies related to the spread of infection (testing, contact tracing, and mask policy) were used. These 11 indicators are presented in Table 1.

The OXCGRT uses eight indicators for containment and closure policies and indicators for public information campaigns to calculate the stringency of containment policies that restrict people’s actions and designates it as a ‘Stringency Index’. Therefore, the Stringency index was used to compare the stringency of national quarantine policies.

All data can be downloaded from the ‘Our World in Data’ homepage, which provides information related to COVID-19 in an easy-to-understand format. The data were retrieved on 4 January 2022.

Of more than 180 countries, 36 OECD member countries with comparable economies and political maturity were targeted for the analysis. As of 16 August 2022, the OECD member countries were 38, but Colombia, which joined in 2020, and Costa Rica, which joined in 2021, were excluded from the analysis due to data stability.

The purpose of this study was to identify the impact of government intervention on COVID-19. In other words, it aimed to find the optimal countermeasures to reduce the harm (deaths, confirmed cases) caused by COVID-19. To this end, the scope of this study targeted OECD member countries to reduce the differences in response policies that occurred due to socioeconomic factors. Here, we analyzed OECD countries with similar levels of economic power and political maturity, and we assumed that quarantine policies for government intervention were similar.

The period for the data was the 5th week of December 2021, when the number of new COVID-19 cases had exploded globally since the Omicron variant was first detected in Botswana on 11 November 2021. Considering the different timing for the quarantine policies on the outbreak of COVID-19, the COVID-19 outbreak trend index data and quarantine policy indicators for the second to fifth weeks of December 2021 were used for analysis.

### 2.2. Analysis

The analyses were divided into the following steps: basic data construction, COVID-19 outbreak trend analysis, classifying country groups according to the outbreak trend, comparative analysis of quarantine policies among groups, and analysis of factors associated with the low risk of infection. 

First, the number of new deaths per million population, number of new confirmed cases, 11 indicators for quarantine policies, and the Stringency Index data, which were established daily for each country, were averaged arithmetically, and converted to weekly data. Since the quarantine policy indicators were evaluated on an ordinary scale, the arithmetic average was rounded to one decimal place and converted to an integer.

Based on the average daily number of new deaths and newly confirmed cases per million population in the fifth week of December 2021 and the average rate of change for the previous four weeks (from the second week of December to the fifth week of December), a graph of the trends of new deaths and confirmed cases of COVID-19 was created. In the graph of incidence trend, quadrants were created based on the daily average number of new deaths and confirmed cases per million population and the average increase and decrease in the rate of 0% for the 36 countries to classify the country groups. A risk score for each group was determined based on the analysis of the new deaths and confirmed cases (Table 2), and the final risk score for each country was calculated by weighting the two scores. The countries were divided into the COVID-19 low-risk, medium-risk, and high-risk groups based on the final risk score.

The stringency of the groups was confirmed by comparing the Stringency Indexes by country groups. In addition, cross-tabulation and chi-square tests were used to compare the groups based on the 11 indicators for quarantine policies to determine any statistically significant differences. Multivariate logistic regression was used to determine the quarantine policy indicators that differentiated the low- from the high-risk group.

## 3. Results

### 3.1. Result of Analysis of COVID-19 Outbreak Trend by Country

#### 3.1.1. Trend of New Deaths

Figure 1 shows the (matrix) analysis results considering the daily average number of new deaths per million population in 36 OECD countries and the average rate of change for the previous four weeks. The matrix was divided into low-risk, medium-risk, and high-risk groups based on the average daily number of new deaths (3.04/million) and the average change rate of 0%. 

As shown in Figure 1, the low-risk group comprised countries that had a relatively low and decreasing death rate. These 10 countries included New Zealand, The Netherlands, and Israel. The medium-risk group included countries with higher than average death rates but the rate of change kept decreasing such as Hungary, Slovakia, and the Czech Republic and those with relatively low death rates that showed a recent increase such as Korea, Norway, and Sweden. The high-risk group comprised countries with high death rates that increased by approximately 5% such as Poland and Lithuania.

#### 3.1.2. Trend of New Confirmed Cases

The number of new confirmed cases, along with the number of new deaths, can provide insights into the trend of COVID-19. Figure 2 shows the (matrix) trends based on the number of new confirmed cases in 36 OECD countries and the average rate of change over the last four weeks. 

The matrix was divided into low-, medium-, and high-risk sections based on the average daily number of new confirmed cases (1042.77/million) and the average change rate of 0%. As shown in Figure 2, the low-risk group included 12 countries such as Korea, New Zealand, and Germany, where the number of new confirmed cases was low and decreased. The medium-risk group included 12 countries such as Australia, New Zealand, and Germany, where the number of new cases was low, but the rate of change increased. The high-risk group included 12 countries such as Denmark, the UK, and France, where the number of new confirmed cases was high, and the number of confirmed cases was increasing.

#### 3.1.3. Country Groups Based on the Trends of Occurrence

As shown in Table 2, the countries were classified based on the trends of new deaths and confirmed cases and assigned scores. Based on these, the groups were finally classified into the low-, medium-, and high-risk groups. For the high-risk group, the rates of new deaths and confirmed cases were above the risk level, and one of them was high; the high-risk group included 11 countries such as Denmark and Italy. The medium-risk group included 10 countries, such as Australia and Canada, where the rates of new deaths and confirmed cases were high or the two had contrasting levels. The low-risk group included 15 countries, such as Korea and New Zealand, where the rates of new deaths and confirmed cases were below the risk level, and one was designated as low (Table 3).

### 3.2. Results of Comparative Analysis of the Quarantine Policies by Country

#### 3.2.1. Comparative Analysis of Governments’ Stringency Index

Figure 3 and Table 4 show the comparison of the countries based on the Stringency Index for COVID-19 for the four weeks before the analysis. Since all three groups had extreme values, it was necessary to compare the Stringency Indexes using the median. The median Stringency Index was 51.3 points for the low-risk group, 45.4 points for the medium-risk group, and 47.7 points for the high-risk group. 

Meanwhile, although the Stringency Indexes of France, Greece, and Italy among the high-risk group were significantly high at 72.2, 80.1, and 76.9, respectively, the number of new confirmed cases increased rapidly, suggesting that a detailed analysis of the quarantine policy indicators was necessary.

#### 3.2.2. Comparative Analysis of Quarantine Policy Indicators

In this section, cross-tabulation and chi-square tests were performed to compare the policies implemented by the country groups according to the level of risk. As shown in Table 5, the groups showed statistically significant differences for 9 of 11 indicators. There were no significant differences in mobility restrictions and contact tracing implementations between the country groups.

The proportion of countries requiring school closures (ordinal points 2 and 3) in the low-risk group was 41.6%, which was much higher than those for the medium-risk (15.0%) and high-risk (27.3%) groups. As the most powerful measure, 13.3% of the low-risk group requested the closure of all schools (ordinal point 3), compared with 0.0% for the medium-risk and high-risk groups.

Regarding workplace closures, the proportion of countries requiring closure (working from home) of all workplaces except essential workplaces (grocery stores, hospitals, etc.) (ordinal point 3) in the low-risk group was 20.0%, which was much higher than those for medium-risk group (2.5%) and high-risk group (11.4%). Regarding the proportion of countries requiring some closure (ordinal point 2), the high-risk group (65.9%) showed a higher value than the low-risk (38.3%) and the medium-risk (27.5%) groups. 

The proportion of countries requiring the cancellation of public events (ordinal point 3) in the high-risk group was 81.8%, which was much higher than those of the low-risk (40.0%) and medium-risk (40.0%) groups. Most groups allowed public gatherings of less than 100 people (ordinal points 3, 4), but the medium risk group (47.5%) had the highest proportion of countries allowing gatherings of fewer than ten people (ordinal point 4).

Regarding the stay-home requirements, it was found that 51.6% of the low-risk group countries requested people to refrain from going out (ordinal point 1) or banned unnecessary outings (excluding grocery shopping, etc.) (ordinal point 2). The proportions were 25.0% and 27.3% for the medium- and high-risk groups, respectively, which were lower than that of the low-risk group.

Regarding close public transport, 20% of the low-risk countries pursued a strong public transport closure policy (ordinal point 2), compared with 27.3% of the high-risk group. Regarding international travel controls, 26.7% of the low-risk countries implemented a complete border closure policy (ordinal point 4), compared with 0.0% of the medium- and high-risk groups. 

Regarding the testing policy for COVID-19, the percentage of countries offering public testing (ordinal point 2), including asymptomatic patients, was 53.3% in the low-risk group, 40.0% in the medium-risk group, and 81.8% in the high-risk group.

Regarding facial coverings, 6.7% of the low-risk countries enforced always wearing a mask outside the home (ordinal point 4), regardless of location or presence of others, whereas relatively low percentages of the medium-risk (0.0%) and high-risk (2.3%) groups enforced strong measures.

### 3.3. Analysis of Multivariate Factors Influencing Classification into a Low-Risk Group

To identify the characteristics of the low-risk group, multivariate logistic regression of the quarantine policy variables (9 variables such as school closure, workplace closure, and gathering restrictions) was performed. The low-, medium-, and high-risk groups showed significant differences. Table 6 shows the results of the best-fitted model using hierarchical backward elimination from the full model that considered all variables.

Multivariate logistic regression analysis was conducted with the risk classification as the dependent variable and stringency of school closure, business closure, public event, public gathering, staying at home, mask, public transportation, international travel, and test targets as the independent variables. Workplace closure, public events restrictions, public gathering restrictions, staying at home, and test targets were selected as independent variables for the best-fitted model. The Hosmer & Lemeshow test was used to test the model’s fit, and the chi-square value indicated the degree of agreement between the actual value of the dependent variable and the predicted value by the model. The significance level was 0.29, indicating that the model of this study was statistically suitable because the null hypothesis was not rejected.

In the best-fitted model, the indicators that had a significant effect (*p* < 0.05) on the classification of the low-risk group were workplace closure, public event restriction, staying at home, and test targets. Countries that had recently implemented strict policies for workplace closure (OR = 2.96) and staying at home (OR = 1.86) were more likely to belong to the low-risk group. On the other hand, countries that had recently implemented strict policies for public events (OR = 0.15) and test targets (OR = 0.31) were less likely to belong to the low-risk group. In other words, countries that had recently implemented more lenient policies for public events and test targets were more likely to belong to the low-risk group.

The test targets were evaluated using an ordinary scale (0, 1, 2): 0 indicated that tests were performed when symptoms of COVID-19 manifested, and certain criteria such as overseas return, hospitalization, and key personnel were met; 1 indicated that tests were performed when symptoms of COVID-19 manifested; and 2 indicated that public testing was performed even for asymptomatic individuals. After incorporating the model analysis results with the definitions for each category, countries with a reduced range of COVID-19 tests were more likely to be classified as low-risk.

The public event restrictions were also evaluated using an ordinary scale (0, 1, 2): 0 indicated no relevant policies; 1 indicated recommendations on the cancellation of public events; and 2 indicated requiring cancellation according to laws and regulations. Countries that had implemented highly mandatory policies for the cancellation of public events were less likely to be classified as low-risk.

## 4. Discussion

Based on the number of new deaths and number of new confirmed cases as of the fifth week of December 2021, when the omicron variant spread globally, 36 countries were classified into the low-, medium-, and high-risk groups. Of the 11 quarantine policy indicators, the restriction of internal movement and contact tracing did not influence the classification of country groups according to the extent of the COVID-19 outbreak. 

Although contact tracing is an important central public health response to an infectious disease outbreak during the early stages of an outbreak, the system can be overwhelmed when the influx of infections is rapid [19]. In this study, the scope of contact tracing (limited or comprehensive) did not seem to have a significant relationship with the level of the COVID-19 outbreak when the infection spread rapidly in a country where the epidemic had already progressed to a large extent. The degree of strictness of policies restricting movement between regions within the same country did not also show a significant relationship with the extent of the COVID-19 outbreak, which is consistent with the results of a previous study [14]. In other words, if it is estimated that community transmission has already occurred, policies that reduce interactions between individuals rather than control by the community unit are associated with new deaths and confirmed cases.

As lockdown measures become stricter, the number of new confirmed cases decreases, and institutional power plays a greater role than cultural and economic factors in improving the outcomes of lockdown measures [20]. Containment and closure measures are essential to contain the epidemic [6], and strict lockdown measures have great potential to prevent or slow the spread of COVID-19 [21,22]. According to previous major studies, stricter policies for school and workplace closure were associated with a better reduction of the spread of COVID-19, and the staying at home policy did not have a significant effect on the infection transmission rates [7,14].

Countries that increased the stringency of the stay-at-home and workplace closures policies were likely to be classified into the low-risk group, but school closure did not significantly influence the classification into the low-risk group. Because the methodologies of this study and previous studies are different, the reason for this cannot be established. It may be due to the differences in the period analyzed. While this study analyzed cases within a short period (4 weeks) during which the number of infections caused by a variant exploded, previous studies analyzed cases over several months from the initial outbreak of an epidemic. Therefore, if the influx of infection by variants occurs at a rapid rate during a pandemic, it is important to strengthen control over the scope of daily life, such as shutting down all workplaces except essential workplaces (grocery stores, hospitals, etc.) or requiring working from home and prohibiting unnecessary outing.

Regarding the test target policies, it was found that stringent testing requirements were associated with a greater likelihood of being classified as low-risk. When several new cases were confirmed every day, such as during the period this study was conducted, countries actively conducted COVID-19 testing, and the number of confirmed cases increased rapidly. It is known that testing regardless of symptoms was related to fewer infections and deaths [23] than testing only when there were symptoms of COVID-19 during the early stages of the pandemic. In other words, it is effective to conduct a wide range of COVID-19 tests during the early stages of a pandemic to control the spread of infectious diseases, but if the infection has already spread to the community on a large scale, the number of confirmed cases and deaths seems to correspond to the number of tests.

Lastly, it was found that countries that implemented highly mandatory policies for the cancellation of public events were less likely to be classified as low-risk. This contrasts with the results of previous studies. The cancellation of public events in Korea and Japan, classified as the low-risk group, was assigned one ordinal point, indicating that their governments recommended the cancellation of official events. Regarding the official announcement at that time by Korea, separate plans were presented for subdivisions of events including large-scale events and assemblies and events related to public affairs and essential business activities, expos, and international conferences, among others. Instead of forcibly demanding cancellation, events started with quarantine measures according to the characteristics of the event, such as the number of people and area, and cancellation was the last resort [24]. For Japan, venues were classified into action zones at the regional level based on the infection and medical conditions, and limits and acceptance rates according to the level assigned to the action zone were applied to the events. In addition, a safety plan to prevent infection was established in advance for large events [25].

This study aimed to investigate the impact of government intervention to contain the spread of infection. From this point of view, economic policy indicators (income support, debt relief for households) and vaccine policy indicators (vaccine prioritization, etc.) are considered to be indirect indicators to contain the spread of infection. Although economic support such as income preservation and debt relief can have some effect on reducing the social activities of the people, it is thought that it cannot directly affect the spread of infection like policies such as movement restrictions and school closures.

In the same context, it was judged that vaccine policy does not have a direct effect on the suppression of the spread of the infection; thus, based on these reasons, this paper excluded economic policy indicators and vaccine policy indicators from the analysis.

Based on the cases of Korea and Japan, the less strict policies related to public events in the low-risk countries were probably due to the precision of the recommended guidelines by the governments and the degree of compliance. It is necessary to conduct in-depth research on this in the future.

## 5. Conclusions

The WHO urged countries to actively respond to the global spread of COVID-19, which is not just a public health crisis, but a crisis that affects all fields, including politics, economy, society, and culture [26]. From a public health point of view, the Find, Test, Tracing, Isolation, Support system and containment and closure policies to prevent the spread of infectious diseases play a very important role. Quarantine policies to prevent the spread of infectious diseases were implemented in many countries; however, the levels and trends of infection by country were markedly different. In this study, we analyzed the differences in quarantine policies according to the level of infection by country. Based on this analysis, we derived a quarantine policy with a high impact on risk reduction. The rates of deaths and confirmed cases were used to classify 36 OECD member countries into risk groups, and the implementations of the quarantine policies by the groups were analyzed to determine significant differences. 

The groups showed significant differences for 9 out of 11 indicators included in the Stringency Index related to the epidemic prevention and control policy. Four indicators significantly influenced the allocation of countries to the low-risk group. Countries with strict policies related to workplace closures and staying at home, which significantly affected the low-risk group, were more likely to belong to the low-risk group. Countries included in the medium-risk and high-risk groups may reduce their risk by promoting quarantine policies focusing on indicators that significantly impacted the low-risk countries.

According to the results of this study, the quarantine policies to be prioritized differ with the characteristics of the epidemic. If a pandemic situation is prolonged, implementing a selective quarantine policy based on individual self-quarantine rather than government-led uniform quarantine measures by comprehensively considering the characteristics of each period will be more effective.

The limitations of this study are as follows. First, the OxCGRT dataset used as the quarantine policy data in this study only recorded the severity based on the government policy announcement, and the degree of policy compliance could not be measured. Additionally, detailed policy characteristics could not be incorporated due to the limitations of the ordinal approach [27]. Therefore, it is necessary to further analyze the risk of COVID-19 according to the level of enforcement for each policy and the degree of compliance in the future and compare the findings with the results of this study. There is also an issue with the timing of the analysis. This study analyzed cases during the fifth week of December 2021. At this time the Omicron variant, which infected more easily and spread faster than the previous Delta variant, spread globally after COVID-19 was declared an international public health emergency by the WHO. On the other hand, existing studies focused on the early stages of the COVID-19 outbreak. It is necessary to research effective quarantine policies according to the predominant variants, epidemic period, and rate of spread.

## Figures and Tables

**Figure 1 ijerph-19-16070-f001:**
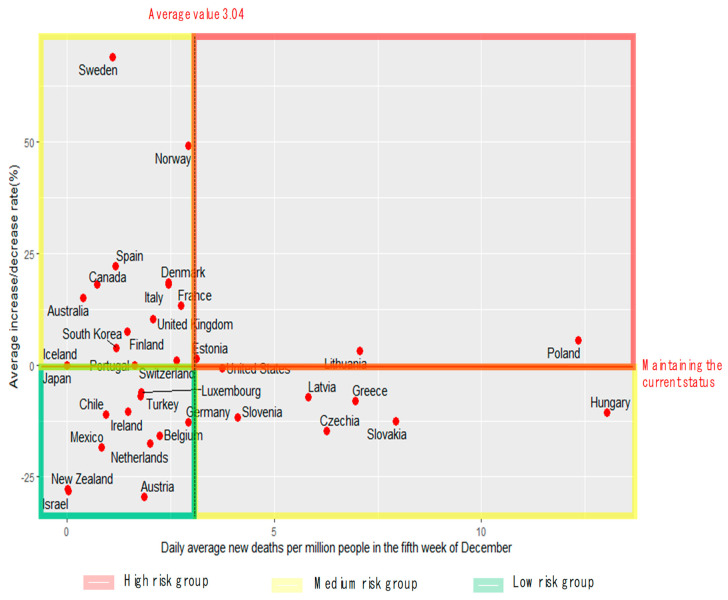
Average daily number of new deaths during the fifth week of December vs. the average rate of change for four weeks.

**Figure 2 ijerph-19-16070-f002:**
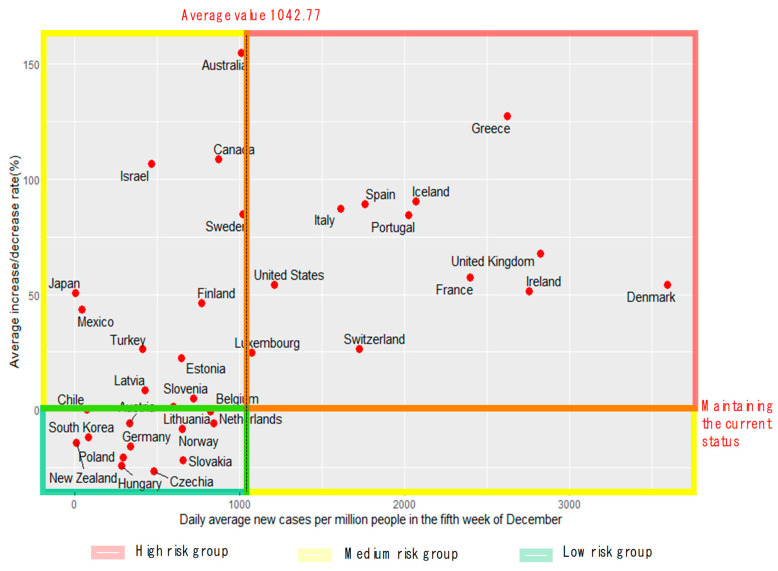
Average daily number of new confirmed cases during the 5th week of December vs. the average rate of change for four weeks.

**Figure 3 ijerph-19-16070-f003:**
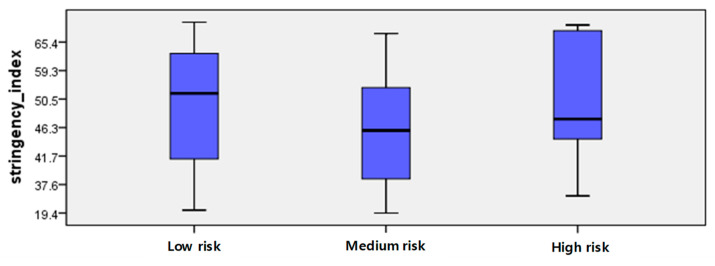
Distribution chart of stringency index of the groups.

**Table 1 ijerph-19-16070-t001:** Governments response indicators.

ID	Name	Description	Measurement
**Containment and closure**			
C1	School closing	Record closing of schools and universities	Ordinal scale
C2	Workplace closing	Record closing of workplaces	Ordinal scale
C3	cancel public events	Record canceling public events	Ordinal scale
C4	Restrictions on gathering size	Record the cut-off size for bans on gatherings	Ordinal scale
C5	Close public transport	Record closing of public transport	Ordinal scale
C6	Stay-at-home requirements	Record orders to “shelter-in-place” and otherwise confinement to the home	Ordinal scale
C7	Restrictions on internal movement	Record restrictions on internal movement	Ordinal scale
C8	Restrictions on international travel	Record restrictions on international travel	Ordinal scale
**Health measures**			
H1	Testing policy	Who can get tested?	Ordinal scale
H2	Contact tracing	Are governments doing contact tracing?	Ordinal scale
H3	Facial coverings	Record policies on the use of facial coverings outside the home	Ordinal scale

**Table 2 ijerph-19-16070-t002:** National group separation criteria.

New Deaths per Million (a)		New Cases per Million (b)		Final Group Classification (a × b)	
Group	Risk score	Group	Risk score	Group	Range of risk score
Low risk	1	Low risk	1	Low risk	1~2
Medium risk	2	Medium risk	2	Medium risk	3~5
High risk	3	High risk	3	High risk	6~9

**Table 3 ijerph-19-16070-t003:** List of countries by group according to risk level.

National Group	Country Name (36 Countries)
Low-risk group (15 countries)	Austria, Belgium, Chile, Czechia, Germany, Hungary, Israel, Japan, Mexico, The Netherlands, Norway, New Zealand, Slovakia, South Korea, Turkey
Medium-risk group (10 countries)	Australia, Canada, Finland, Iceland, Ireland, Latvia, Luxembourg, Poland, Slovenia, Sweden
High-risk group (11 countries)	Denmark, Estonia, France, Greece, Italy, Lithuania, Portugal, Spain, Switzerland, United Kingdom, United States

**Table 4 ijerph-19-16070-t004:** Stringency index by country.

	Low-Risk Group		Medium-Risk Group		High-Risk Group	
	Country	Stringency Index	Country	Stringency Index	Country	Stringency Index
Max.	Germany	84.3	Canada	69.6	Greece	80.1
Min.	Hungary	25.0	Switzerland	19.4	Estonia	34.3
Med.	-	51.3	-	45.4	-	47.7

**Table 5 ijerph-19-16070-t005:** Relationship between groups and government response policies.

Policy Indicators	Classification by Ordinal Point	National Group N(%)			χ〖〗LSUP〖2〗(p)
		Low Risk	Medium Risk	High Risk	
school closures	0—no measures	14 (23.3)	8 (20.0)	8 (18.2)	18.260 (0.006)
	1—recommend closing	21 (35.0)	26 (65.0)	24 (54.5)	
	2—require closing only some levels or categories	17 (28.3)	6 (15.0)	12 (27.3)	
	3—require the closing of all levels	8 (13.3)	0 (0.0)	0 (0.0)	
workplace closures	0—no measures	4 (6.7)	4 (10.0)	0 (0.0)	25.596 (0.000)
	1—recommend closing	21 (35.0)	24 (60.0)	10 (22.7)	
	2—require closing for some sectors or categories of workers	23 (38.3)	11 (27.5)	29 (65.9)	
	3—require closing all but essential workplaces	12 (20.0)	1 (2.5)	5 (11.4)	
cancel public events	0—no measures	3 (5.0)	4 (10.0)	0 (0.0)	23.325 (0.000)
	1—recommend canceling	33 (55.0)	20 (50.0)	8 (18.2)	
	2—require canceling	24 (40.0)	16 (40.0)	36 (81.8)	
restriction gatherings	0—no measures	8 (13.3)	6 (15.0)	0 (0.0)	22.625 (0.004)
	1—restrictions on very large gatherings	4 (6.7)	0 (0.0)	7 (15.9)	
	2—restrictions on gatherings between 100–1000 people	4 (6.7)	1 (2.5)	6 (13.6)	
	3—restrictions on gatherings between 10–100 people	24 (40.0)	14 (35.0)	22 (50.0)	
	4—restrictions on gatherings of less than 10 people	20 (33.3)	19 (47.5)	9 (20.5)	
stay home requirements	0—no measures	29 (48.3)	30 (75.0)	32 (72.7)	17.284 (0.002)
	1—recommend not leaving the house	20 (33.3)	10 (25.0)	4 (9.1)	
	2—require not leaving the house with exceptions for daily exercise, grocery shopping, and ‘essential’ trips	11 (18.3)	0 (0.0)	8 (18.2)	
	3—require not leaving the house with minimal exceptions	0 (0.0)	0 (0.0)	0 (0.0)	
close public transport	0—no measures	40 (66.7)	24 (60)	24 (54.5)	18.803 (0.001)
	1—recommend closing	8 (13.3)	16 (40)	8 (18.2)	
	2—require closing	12 (20.0)	0 (0.0)	12 (27.3)	
international travel controls	0—no measures	0 (0.0)	0 (0.0)	0 (0.0)	44.917 (0.000)
	1—screening	10 (16.7)	0 (0.0)	0 (0.0)	
	2—quarantine arrivals from high-risk regions	12 (20.0)	12 (30.0)	16 (36.4)	
	3—ban on high-risk regions	22 (36.7)	28 (70.0)	28 (63.6)	
	4—total border closure	16 (26.7)	0 (0.0)	0 (0.0)	
testing policy	0—only those who both (a) have symptoms AND (b) meet specific criteria	12 (20.0)	2 (5.0)	0 (0.0)	28.225 (0.000)
	1—testing of anyone showing COVID-19 symptoms	16 (26.7)	22 (55.0)	8 (18.2)	
	2—open public testing	32 (53.3)	16 (40.0)	36 (81.8)	
restriction internal movements	0—no measures	36 (60.0)	32 (80.0)	26 (58.1)	7.918 (0.095)
	1—recommend movement restriction	8 (13.3)	0 (0.0)	5 (11.6)	
	2—restrict movement	16 (26.7)	8 (20.0)	13 (30.2)	
facial coverings	0—no measures	0 (0.0)	0 (0.0)	0 (0.0)	24.540 (0.002)
	1—recommended	4 (6.7)	4 (10.0)	0 (0.0)	
	2—required in some specified shared/public spaces outside the home with other people present or in some situations when social distancing is not possible	28 (46.7)	10 (25.0)	14 (31.8)	
	3—required in all shared/public spaces outside the home with other people present or in all situations when social distancing is not possible	24 (40.0)	22 (55.0)	29 (65.9)	
	4—required outside the home at all times, regardless of location or presence of other people	4 (6.7)	0 (0.0)	1 (2.3)	
contact tracing	0—no contact tracing	4 (6.7)	4 (10.0)	0 (0.0)	8.575 (0.073)
	1—limited contact tracing is not done for all cases	16 (26.7)	18 (45.0)	18 (40.9)	
	2—comprehensive contact tracing—done for all cases	40 (66.7)	18 (45.0)	26 (59.1)	
Total		60	40	44	

**Table 6 ijerph-19-16070-t006:** Multiple logistic regression of good group on each response policy level.

Variable	Full Model			Best-Fitted Model *			
	Estimate	Std.	Pr (>|z|)	Estimate	Std.	Pr (>|z|)	OR (95% CI)
(Intercept)	2.78	1.32	0.03	2.38	0.88	0.01	10.83(2.04–65.09)
school closures	0.10	0.40	0.79	-	-	-	-
workplace closures	1.12	0.40	0.01	1.08	0.37	0.00	2.96(1.49–6.34)
cancel public events	−1.73	0.49	0.00	−1.91	0.47	0.00	0.15(0.06–0.36)
restriction gatherings	0.25	0.21	0.23	0.29	0.18	0.12	1.33(0.93–1.93)
stay home requirements	0.76	0.42	0.07	0.62	0.28	0.03	1.86(1.08–3.27)
facial coverings	−0.07	0.30	0.81	-	-	-	-
close public transport	−0.38	0.34	0.27	-	-	-	-
International travel controls	−0.12	0.29	0.69	-	-	-	-
testing policy	−1.19	0.39	0.00	−1.16	0.36	0.00	0.31(0.15–0.62)

* Hosmer & Lemeshowl χ^〖2〗  = 9.69 (df = 8, *p*-value = 0.29).

## Data Availability

Not applicable.

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
