# Peer review of "Impact of Government Intervention in Response to Coronavirus Disease 2019"

_ijerph, 2022, doi:10.3390/ijerph192316070_

Round 1

Reviewer 1 Report

Please refer to the attached document.

Author Response

I sincerely appreciate the reviewer’s comments on and contributions toward the manuscript. Although I undertook utmost care with the manuscript’s content before submission, I have found your comments to be very useful and valid. Following are the revisions made in response:

Point 1: 

 Lines 111-119 say, “The different policy measures were tracked since January 1, 2020, for more than 180 countries and categorized under 21 indicators, such as school closures, travel restrictions, and vaccination policy. These policies were evaluated on a scale to reflect the extent of government action, and scores were aggregated into a suite of policy indices. The policy indicators included: eight on lockdown and closure policies (school closure, movement restriction, etc.), two on economic policies (income support, debt relief for households), seven on health system policies (testing policy, contact tracing, etc.), and four on vaccine policies (vaccine prioritization, etc.).” 
This manuscript would be significantly improved if the analyses included the economic and vaccine policies. How much would the economic and vaccine policies influence the impact of quarantine policies on government intervention in a pandemic like COVID-19? Would the conclusion based on quarantine policies in different risk groups remain the same or not?

Response 1

We thank the reviewer for this input. We believe that the investigation of economic and vaccine policies falls beyond the scope of the current manuscript. We are preparing subsequent analyses to account for these parameters. As per the comments, we have supplemented the text in lines 420-429.

This study aimed to investigate the impact of government intervention to contain the spread of infection. From this point of view, we consider economic policy indicators (income support, debt relief for households) and vaccine policy indicators (vaccine prioritization, etc.) to be indirect indicators to contain the spread of infection. Although economic support such as income preservation and debt relief can have some effect on reducing the social activities of the people, we believe that it cannot directly affect the spread of infection in contrast to policies such as movement restrictions and school closures. 
In the same context, we judged that vaccine policy does not have a direct effect on the suppression of the spread of the infection. Based on these reasons, this paper excluded economic policy indicators and vaccine policy indicators from the analysis. In the future, we plan to conduct research including various government policies along with policies to contain the spread of infection. 

Reviewer 2 Report

This study examines the quarantine policies according to the level of risk of COVID-19 and their strictness in each country and then determined the most effective intervention in reducing risks. Through the analysis, this study identifies the types of quarantine policies and their effectiveness in reducing the risk of infectious diseases.

The authors proposed a risk score that was determined based on the analysis of the new deaths and confirmed cases. They used the score for classifying the 36 countries into 3 groups. The analysis results highly depend on the classification result because they compared differences among these groups. However, this study lacks the following two points:

1. Literature review on COVID-19 risk scores. 

It is strongly recommended to add a literature review on various risk scores that are relevant for classifying OECD countries where they elevate the effect of policy intervention. The review may include 1) a definition of risk score, 2) factors used for constructing the scores, and 3) the rationale for including the factors when the risk scores are used for classifying the 36 countries.

For example, the EU published a document about the INFORM COVID-19 Risk Index that includes various dimensions for analyzing COVID-19 risks. 

2. Examining the relevance of the COVID-19 risk score that they are employing

It is strongly recommended that authors additionally perform an analysis of the relevance of the COVID-19 risk score that they proposed. The current-proposed score uses only the number of confirmed cases and new death alone. 

The score does not take into account any socio-economic, such as economic inequality, and institutional factors, such as governance capacity, of the country. Given that the effect of policy intervention is closely related to socio-economic and governance factors, it is suggested that authors examine whether the COVID-19 risk score they are proposing can effectively evaluate the risks and classify the 36 countries in a relevant manner

Author Response

I sincerely appreciate the reviewer’s comments on and contributions toward the manuscript. Although I undertook utmost care with the manuscript’s content before submission, I have found your comments to be very useful and valid. Following are the revisions made in response:

Point 1: 
This study examines the quarantine policies according to the level of risk of COVID-19 and their strictness in each country and then determined the most effective intervention in reducing risks. Through the analysis, this study identifies the types of quarantine policies and their effectiveness in reducing the risk of infectious diseases.

The authors proposed a risk score that was determined based on the analysis of the new deaths and confirmed cases. They used the score for classifying the 36 countries into 3 groups. The analysis results highly depend on the classification result because they compared differences among these groups. However, this study lacks the following two points:

1. Literature review on COVID-19 risk scores. 

It is strongly recommended to add a literature review on various risk scores that are relevant for classifying OECD countries where they elevate the effect of policy intervention. The review may include 1) a definition of risk score, 2) factors used for constructing the scores, and 3) the rationale for including the factors when the risk scores are used for classifying the 36 countries.
For example, the EU published a document about the INFORM COVID-19 Risk Index that includes various dimensions for analyzing COVID-19 risks.

Response 1

We thank the reviewer for their valuable comments. As per your comments, we supplemented lines 95-107.
Additional references [18] were also added.

Referring to the recommendations, we reviewed the INFORM COVID-19 Risk Index document and added it to the ‘Literature Review’ section of the paper. As a result of our review, the INFORM COVID-19 Risk Index is a composite index to identify countries at risk due to COVID-19 in need of additional international assistance, and, as you mentioned, country-by-country comparisons are possible. However, it is divided into Hazard & Exposure, Vulnerability, and Lack of coping capacity, and new deaths and new confirmed cases are not reflected as indicators to identify the current level of the COVID-19 outbreak. The purpose of this study is to identify policies related to the suppression of the spread of infection that affect the level of the COVID-19 outbreak (risk level) at a certain point in time. 
Since this is different from the calculation purpose of the INFORM COVID-19 Risk Index, it is judged that it will be difficult to reflect it in the research methodology of this paper. 

Point 2: 
Examining the relevance of the COVID-19 risk score that they are employing

It is strongly recommended that authors additionally perform an analysis of the relevance of the COVID-19 risk score that they proposed. The current-proposed score uses only the number of confirmed cases and new death alone. 

The score does not take into account any socio-economic, such as economic inequality, and institutional factors, such as governance capacity, of the country. Given that the effect of policy intervention is closely related to socio-economic and governance factors, it is suggested that authors examine whether the COVID-19 risk score they are proposing can effectively evaluate the risks and classify the 36 countries in a relevant manner

Response 2

As the reviewer’s comments, we have supplemented the text in lines 155-161. To address their point, we approached the current study from a simplified point of view of government-instituted policies among similarly governed countries. Although spanning a range, we considered OECD member countries to have sufficiently equivalent level of response policies. For the scope of this study, we have opted to limit our analysis to the factors listed in the study and are planning future analysis incorporating additional risk factors such as economic and vaccination policies (a point we also answered in response to Reviewer 1).

The purpose of this study was to infer the impact of government intervention on COVID-19. In other words, it aimed to find the optimal countermeasures to reduce the damage (deaths, confirmed cases) caused by COVID-19. To this end, the scope of this study targeted OECD member countries to reduce the differences in response policies that occurred due to socioeconomic factors. Here, we analyzed OECD countries with similar levels of economic power and political maturity, and we assumed that quarantine policies for government intervention were similar. 

Round 2

Reviewer 1 Report

I believe the manuscript has been sufficiently improved to warrant publication in IJERPH.

Author Response

Thank you very much.

Reviewer 2 Report

I am afraid that the review of the risk score is not thorough at all. I don't think INFORM COVID-19 Risk Index is not applicable to this study but I don't see the reason why the author could easily conclude that their index is better than the INFORM COVID-19 Risk Index. That kind of conclusion can be made after comparing/examining several similar indicators. I don't think that the risk score the author is employing is well justified in the context of the review of the existing score. 

Re. the additions they made "Here, we analyzed OECD countries with similar levels of economic power and political maturity, and we assumed that quarantine policies for government intervention were similar."

What is the based of that assumption? Does this mean the authors consider that the "economic power and political maturity" of the target countries by their definition are similar enough so these factors will not be factors affecting the impact of government intervention? If so, please explain it by evidence.

Author Response

I sincerely appreciate your comments on and contributions to the manuscript. Although I undertook utmost care with the manuscript’s content before submission, I have found your comments to be very useful and valid. Following are point-by-point remarks and revisions made in response to your comments:

Point 1: 
I am afraid that the review of the risk score is not thorough at all. 
I don't think INFORM COVID-19 Risk Index is not applicable to this study 
but I don't see the reason why the author could easily conclude that their index is 
better than the INFORM COVID-19 Risk Index. 
That kind of conclusion can be made after comparing/examining several similar indicators. 
I don't think that the risk score the author is employing is well justified in 
the context of the review of the existing score.

Response

Thank you for this valuable comment. I want to reiterate that we do not believe that our index is better than the INFORM COVID-19 Risk Index, but that it should be used according to the purpose of the analysis, i.e., the analysis of whether structural risk factors that existed before the outbreak affected government response policies. As described in more detail below, the purpose of this index is different from the analysis of the government response policy (the index we used) after the outbreak.
However, I agree with your comment that it is necessary to compare and review similar indicators. We plan on incorporating such comparisons in future research.

As per your comments, we added lines 95-107 and another reference [18].

Referring to your recommendations, we reviewed the INFORM COVID-19 Risk Index document and added it to the ‘Literature Review’ section of the paper. The INFORM COVID-19 Risk Index is a composite index to identify countries at risk due to COVID-19 and in need of additional international assistance, and, as you mentioned, country-by-country comparisons are possible. However, the index is divided into Hazard & Exposure, Vulnerability, and Lack of coping capacity, and new deaths and new confirmed cases are not reflected as indicators to identify the current level of the COVID-19 outbreak. The purpose of this study is to identify policies related to the suppression of the spread of infection that affect the level of the COVID-19 outbreak (risk level) at a certain point in time. 
Since this is different from the purpose of the INFORM COVID-19 Risk Index, we believe that it will be difficult to reflect it in the research methodology of this paper. 

Point 2: 
Re. the additions they made "Here, we analyzed OECD countries with similar levels of economic power and political maturity, and we assumed that quarantine policies for government intervention were similar."

What is the based of that assumption? 
Does this mean the authors consider that the "economic power and political maturity" of the target countries by their definition are similar enough so these factors will not be factors affecting the impact of government intervention? If so, please explain it by evidence.

Response 2

Thank you for asking an interesting question. The 'risk score' presented in this study is simply a score to understand the level of COVID-19 outbreak (damage). It does not represent the risks of economic inequality, socio-economic factors, and infectious diseases associated with the country's ability to govern.
In this study, national groups (risk groups) were classified based on the damage results. Although each country promoted a similar type of quarantine policy, it was assumed that the cause of different damage results was the degree of government intervention.

We think all these indicators should be considered in order to clearly understand the effectiveness of government intervention because economic level, hygiene level, and political maturity affect the damage results. However, it was judged that it was difficult to analyze the effect by finding all of these indicators and considering them as control variables. So the target was limited to OECD member countries with a relatively small difference in economic power and political maturity. However, since the data used in this study recorded the degree of rigor based on policy presentation data, the fact that state compliance (degree of policy compliance) was not reflected in our indicators is currently specified as a limitation in the revised Plaintiff's conclusion. (See lines 471 through 473).

We think further research is needed by improving the areas that acted as limitations in this study.